# Molecular mechanisms of fentanyl mediated β-arrestin biased signaling

**Parker W. de Waal**[1¤], **Jingjing Shi**[2], **Erli You**[2], **Xiaoxi Wang**[2], **Karsten Melcher**[1], **Yi Jiang**[2]*, **H. Eric Xu**[1,2]*, **Bradley M. Dickson**[3]*

**1** Center for Cancer and Cell Biology, Innovation and Integration Program, Van Andel Research Institute, Grand Rapids, Michigan, United States of America, **2** The CAS Key Laboratory of Receptor Research, Shanghai Institute of Materia Medica, Chinese Academy of Sciences, Shanghai, China, **3** Center for Epigenetics, Van Andel Research Institute, Grand Rapids, Michigan, United States of America

¤ Current address: D. E. Shaw Research, New York, New York, United States of America
* yijiang@simm.ac.cn (YJ); eric.xu@simm.ac.cn (HEX); bradley.dickson@vai.org (BMD)

**Data Availability Statement:** Parameters, initial coordinates, and simulation restart files for all systems are available at the fABAMBER github page https://github.com/ParkerdeWaal/fABAMBER. All trajectories are available upon request.

## Abstract

The development of novel analgesics with improved safety profiles to combat the opioid epidemic represents a central question to G protein coupled receptor structural biology and pharmacology: What chemical features dictate G protein or β-arrestin signaling? Here we use adaptively biased molecular dynamics simulations to determine how fentanyl, a potent β-arrestin biased agonist, binds the μ-opioid receptor (μOR). The resulting fentanyl-bound pose provides rational insight into a wealth of historical structure-activity-relationship on its chemical scaffold. Following an in-silico derived hypothesis we found that fentanyl and the synthetic opioid peptide DAMGO require M153 to induce β-arrestin coupling, while M153 was dispensable for G protein coupling. We propose and validate an activation mechanism where the n-aniline ring of fentanyl mediates μOR β-arrestin through a novel M153 "micro-switch" by synthesizing fentanyl-based derivatives that exhibit complete, clinically desirable, G protein biased coupling. Together, these results provide molecular insight into fentanyl mediated β-arrestin biased signaling and a rational framework for further optimization of fentanyl-based analgesics with improved safety profiles.

## Author summary

The global opioid crisis has drawn significant attention to the risks associated with over-use of synthetic opioids. Despite the public attention, and perhaps in-line with the profit-based incentives of the pharmaceutical industry, there is no public structure of mu-opioid receptor bound to fentanyl or fentanyl derivatives. A publicly available structure of the complex would allow open-source development of safer painkillers and synthetic antagonists. Current overdose antidotes, antagonists, require natural products in their synthesis which persists a sizable barrier to market and develop better antidotes. In this work we use advance molecular dynamics techniques to obtain the bound geometry of mu-opioid receptor with fentanyl (and derivatives) and uncovered a novel molecular switch involved in receptor activation. Based on our in-silico structure, we synthesized and tested novel

**Funding:** This work is in part supported by the Van Andel Institute Graduate School (PWdW), the Jay and Betty Van Andel Foundation (KM, BMD, and HEX), and the National Natural Science Foundation (31770796 to YJ), the National Science and Technology Major Project (2018ZX09711002 to YJ); the K.C. Wong Education Foundation (YJ). The funders had no role in study design, data collection and analysis, decision to publish, or preparation of the manuscript.

**Competing interests:** The authors have declared that no competing interests exist.

compounds to validate our predicted structure. Herein we report the bound state of several dangerous fentanyl derivatives and introduce new derivatives with signaling profiles that may lead to lower risk of respiratory depression.

## Introduction

Activation of the μ-opioid receptor (μOR), a class A G protein coupled receptor (GPCR), by opiates leads to clinically desired antinociceptive properties [1], but also presents many unwanted on-target effects including addiction and potentially fatal respiratory depression. Opiates represent the largest class of prescribed drugs for the management of severe pain [2], and their subsequent abuse has led to the fastest growing public health epidemic in United States history with the Centers for Disease Control reporting a ten-fold increase in lethal overdoses from 2013 to 2017 [3]. Driven primarily by fentanyl, a highly potent and easily synthesized synthetic opiate [4], the unprecedented rate of opioid overdoses has prompted intensive research on the development of novel analgesics without the risks of standard opiates [5].

Fentanyl, and more commonly prescribed opiate alkaloids such as oxycodone, stimulate desirable antinociceptive activity by interacting with the μOR's orthosteric ligand binding site. Ligand bound μOR, in turn, mediates heterotrimeric Gi/o complex signaling to inhibit cyclic adenosine monophosphate (cAMP) production [6]. Activated μOR can also couple to β-arrestins through recognition of C-terminal tail phosphorylation to promote both receptor internalization and desensitization [7]. This β-arrestin signaling axis has been hypothesized as the source for many unwanted opiate-induced side-effects, as β-arrestin knock-out mice exhibit prolonged morphine induced analgesia, increased resistance to respiratory depression, opiate dependence and tolerance, and constipation [8–11].These studies suggest that decoupling of G protein from arrestin signaling through biased agonism may offer a clinically viable approach to developing safer opioid analgesics. While this hypothesis is based on knock-out models, it remains an open question whether this effect can be replicated by a small molecule.

The development of biased agonists to selectively modulate GPCR-stimulated G protein or arrestin signaling pathways for the μOR has been met with varied success [12–14]. Specifically, oliceridine (TRV130) provides similar antinociceptive activity to morphine while retaining a high potential for dependence and abuse with no respiratory safety advantages over morphine [15]. Another novel opiate, PZM21, was similarly shown to be less potent than morphine while eliciting undesirable respiratory depression and dose-dependent tolerance [16]. This limited success of designing G protein biased μOR agonists is ultimately hindered by a poor mechanistic understanding of how a ligand's chemical structure influences the receptors conformational state to mediate specific signaling pathways.

To gain structural insight into biased signaling, here we sought to understand the mechanism of μOR recognition and activation by fentanyl, a β-arrestin bias agonist. As ligand binding and unbinding events often occur on the timescales unreachable by standard simulations, we introduced a computationally efficient mollified adaptive biasing potential (mABP) [17] in to the graphics processing unit (GPU) enabled AMBER molecular dynamics (MD) engine to accelerate sampling of rare binding events. We simulated binding events for fentanyl, and two of its more potent derivatives carfentanil and lofentanil, to the μOR. These simulations uncovered a common pose within the receptors orthosteric site consistent with a wealth a historical selectivity and structure-activity relationship studies on the fentanyl scaffold. Within this shared pose, fentanyl induced a unique rotameric conformation of M153$^{3.36}$ (Ballesteros-Weinstein numbering), which subsequently displaces W295$^{6.48}$, a conserved "microswitch"

critically important for receptor activation in all class A GPCRs [18]. Mutational studies of M153$^{3.36}$ reveal its role as a specific "microswitch" for μOR β-arrestin signaling as both fentanyl and DAMGO, a synthetic opioid peptide, require M153$^{3.36}$ for β-arrestin but not Gi complex coupling. We further synthesized fentanyl derivatives designed to be unable to modulate the M153$^{3.36}$ β-arrestin microswitch. Consistent with our computationally derived hypothesis, these compounds exhibit no detectable levels of β-arrestin coupling while retaining partial agonist-like Gi coupling. Our study provides a structural basis of fentanyl mediated β-arrestin bias signaling and provides chemical insight for the design of safer fentanyl-based analgesics.

## Results

### Implementation of fABAMBER

Expanding upon our prior work to introduce a computationally efficient, scalable mollified ABP (mABP) scheme in the GROMACS MD engine [19], here we introduced mABP with overfill protection [20] to the graphics processing unit (GPU) accelerated version of pmemd.cuda [21] for AMBER16 [22] which is available for download at Github (https://github.com/ParkerdeWaal/fABAMBER). By minimizing communication between GPU and CPU routines, our mABP implementation, termed as fast-Adaptive-Biasing-AMBER (fABAMBER), introduces nearly no overhead to the native MD engine for small and large simulation systems and provides an approximately 3-fold increase in simulation throughput compared to AMBER's Nonequilibrium Free Energy (NFE) toolkit (Fig 1A). Beyond application to ligand binding presented here, fABAMBER can perform free energy calculations using dihedral, distance, and root-mean-square deviation (RMSD) collective variables (CVs) similar to the AMBER's NFE toolkit and the PLUMED 2 [23] plugin for GROMACS. fABAMBER thus allows users to invoke mABP methodologies while leveraging the full computational efficiency of the AMBER's GPU accelerated pmemd.cuda engine. Additionally, both fABAMBER and fAB-MACS utilize a recent advancement, called overfill protection, that prevents the adaptive bias from driving the system over unrealistic barriers [20]. We discuss the significance of this feature for receptor-ligand binding simulations below.

### Ligand binding to β2AR and μOR

To assess the accuracy of ligand binding pose prediction for GPCRs using fABAMBER, we first simulated binding of carazolol, an inverse agonist, to the β2 adrenergic receptor (β2AR) (S1 Fig) [24]. Ligand binding to β2AR has been previously achieved through brute force computation using ANTON, a specialized computer designed to maximize throughput of unbiased MD steps, and thus serves as a reasonable test-case [25]. Carazolol was initially placed in solvent above β2AR. The ligand-receptor conformation space was projected onto a 2D collective variable (CV) space comprising two root-mean-squared deviations (RMSDs) where the ligand was decomposed into its flexible isopropylamino (CV1) and rigid carbazole moieties (CV2; S1 Fig). The reference points for CV1 and CV2 RMSDs are each a single center-of-geometry (COG), both below the orthosteric site, determined by Cα atoms within TM2/3/7 and TM3/5/6 residues, respectively (S1 Fig). All simulations were run for approximately 2 μs to provide ample sampling of ligand binding and unbinding events. This system setup was applied uniformly to all simulations performed here.

Initial simulations without bias overfill protection resulted in severe structural deformation as ligand pushed through the receptor's secondary structure. In a series of overfill protection calibration experiments (S2 Fig), simulations performed at low (11 and 14 kcal/mol) and high bias (20 kcal/mol) potential fill-limits failed to identify prominent binding events. At a fill-limits of 17 kcal/mol, two of four simulations identifying a common low energy conformation

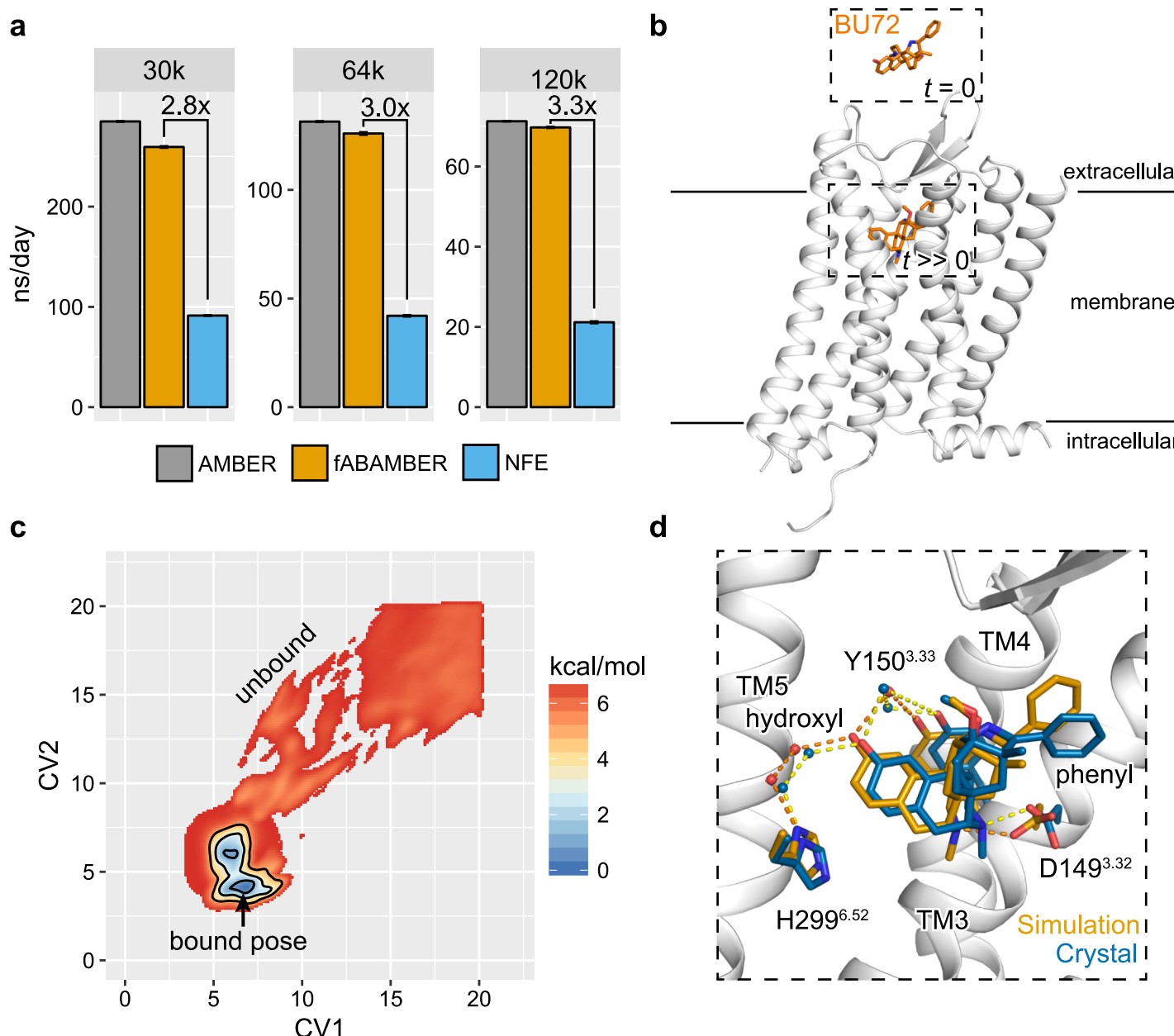

**Fig 1. fABAMBER benchmarking and application to ligand binding. a,** fABAMBER performance compared to AMBER's native GPU enabled simulation engine and NFE toolkit for systems comprising 30k, 64k, and 120k atoms. **b,** Example initial system configuration for ligand binding simulations where the μOR is represented in white and the morphine derived agonist BU72 is shown in orange. **c,** Averaged free energy landscape for BU72 mABP ligand binding simulations. Contours are drawn at 1, 3, and 5 kcal/mol. **d,** Structural overlap of BU72's predicted low-energy conformation (white cartoon and orange sticks) and crystallographic position (PDB Code: 5C1M; blue).

within the receptors orthosteric site which overlapped with the expected crystallographic position with a RMSD less than 0.5 Å (S1 Fig). Importantly, this pose successfully recaptured both crystallographic hydrogen bonding and salt-bridge interactions.

We next simulated the binding of BU72, a potent morphine-derived agonist [26], to the crystal structure of inactive μOR (Fig 1A and S3 Fig) [27]. After a series of calibration experiments for fill-limits were performed similar to that of β2AR, we selected a 15 kcal/mol fill-limit

for all subsequent μOR simulations. Prominent energetic minima were identified within the orthosteric site in two of four simulation replicates (S4 Fig). The representative ligand conformation captured near-atomic level crystallographic interactions between BU72's morphine core and the μOR (Fig 1D). Notably, a salt bridge between the ligand's protonated amine and D149$^{3.32}$ and a water mediated hydrogen bonding network between the distal hydroxyl group to Y150$^{3.33}$ and H299$^{6.52}$. Slight positional variance of the pendant phenyl ring was observed as we simulated the connecting carbon in its sp$^3$ tetrahedral configuration while the crystal structure unexpectedly contained a near-planar sp$^3$ species [28].

As the CVs used above can be viewed as tailored to carazolol and BU72, we also carried out a series of simulations where the COG reference points were switched to explore how CV construction effects sampling efficiency (see S1B Fig and imagine switching colors of the reference points). Across 4 independent simulations for each ligand, no binding events were observed in the carazolol simulations whereas one BU72 simulation identified a local minima within the orthosteric site (S11 Fig). This simulation, however, failed to correctly identify the crystallographic conformation of BU72 suggesting that this alternative CV choice is less efficient than the original construction. Further comment on this pairing is made below.

## Fentanyl binding to the μOR

Encouraged by successful pose identification for carazolol and BU72, we next sought to determine how fentanyl (ED$_{50}$ = 0.011 mg/kg), as well as two more potent derivatives, carfentanil (ED$_{50}$ = 0.00032 mg/kg) and lofentanil (ED$_{50}$ = 0.0006 mg/kg), engage and activate the human μOR (Fig 2A) [29]. Carfentanil and lofentanil both differ from fentanyl through a shared carbomethoxy group at the 4-position of the core piperidine ring while lofentanil has an additional cis-methyl group at the 3-position. Atoms comprising the n-alkyl phenyl ring were assigned to CV1 whereas atoms within piperidine scaffold, propyl-amide, and n-aniline ring were assigned to CV2 without any further experimentation or justification (S3 Fig). Six 2-μs mABP simulation replicates were run for each compound, and a common low-energy conformation emerged across all three ligands (Fig 2B and Fig 2C and S4–S7 Figs and S1–S3 Files). All fentanyl derivatives were oriented in a same latitudinal manner in the orthosteric binding site, forming extensive hydrophobic contacts between their n-alkyl phenyl and propyl-amide groups with transmembrane helices (TM) 2/3 and 3/5/6, respectfully. The n-aniline ring was oriented downwards towards the receptor's hydrophobic core above M153$^{3.36}$. The piperidine ring is angled towards TM3, allowing for a salt bridge to form between its protonated amine and D149$^{3.32}$. The shared 4-carbomethoxy between carfentanil and lofentanil packs against I298$^{6.51}$, V302$^{6.55}$, W320$^{7.35}$, C323$^{7.38}$, I324$^{7.39}$ to provide an additional set of anchoring hydrophobic interactions (Fig 2D and Fig 2E). The adjoining free ketone group points upwards towards the solvent occupied space above the orthosteric site. Lofentanil's 3-cis methyl is sandwiched between I146$^{3.29}$, D149$^{3.32}$, and Y150$^{3.33}$ to increase the hydrophobic interaction surface. Substantial pharmacophore overlap between simulated fentanyl and derivatives to the BU72 bound μOR crystal structure was observed, supporting our computational results (S8 Fig) [28].

## Internal dynamics, affinity, and selectivity

Chemical modifications that both minimize ligand conformational entropy and stabilize productive ligand-receptor contacts have been hypothesized to increase ligand potency [19]. To assess how the carbomethoxy and 3-cis methyl groups of carfentanil and lofentanil affect ligand rigidity, we computed the free energy of rotation around a dihedral defined between the n-aniline and the piperidine ring. Fentanyl's largely flat landscape comprised three energetic

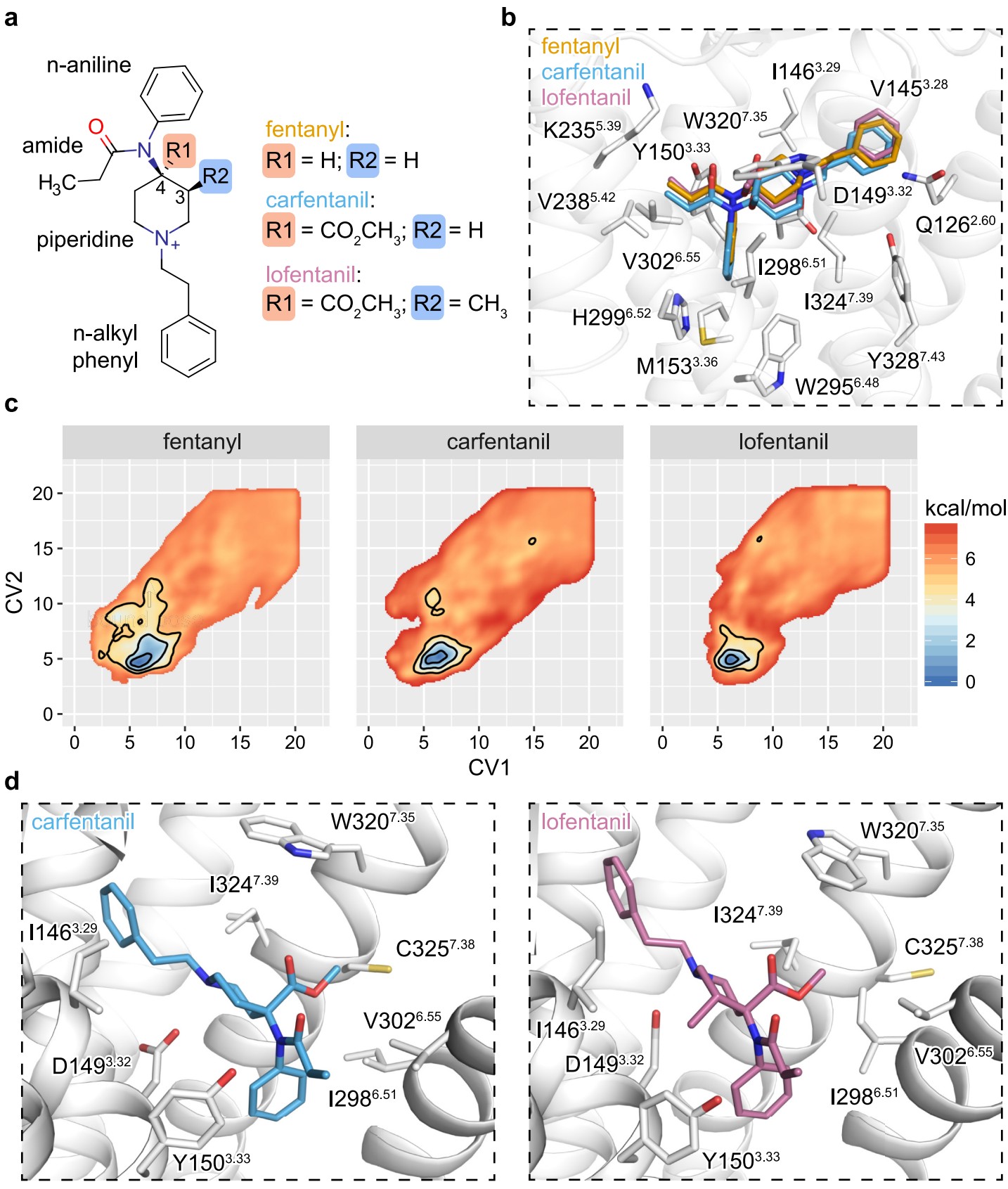

**Fig 2. A common pose for fentanyl binding. a,** Chemical structure of fentanyl, carfentanil, and lofentanil. **b-c,** Structural superposition and sidechain interactions of fentanyl and its derivatives within the μOR orthosteric site and their corresponding averaged free energy landscapes. Contours are drawn at 1, 3, and 5 kcal/mol. **d**, Side-by-side comparison of side-chain interactions mediated through lofentanil's 3-cis methyl group and a common 4-carbomethoxy shared by carfentanil.

minima with no barriers greater than 1.1 kcal/mol (Fig 3A). Two global minima were identified within a shared super basin spanning approximately -2 and 0 radians, separated by a negligible saddle, and a secondary local minimum, S3, located at 2 radians. Addition of a 4-carbomethoxy group (carfentanil) shifted the positions of S1 and S2 closer to -2 and 0 radians respectfully while increasing the barrier separating these two states from 0.1 to 0.9 kcal/mol. The depth of S3 shrank in favor of the primary super basin containing S1 and S2 which also saw a slight increase in its bounding barriers from 1.1 to 1.25 kcal/mol. Secondary addition of a 3-cis methyl group (lofentanil) drastically altered the dihedral landscape in favor of a single global minimum, S2, that is bounded by steep asymmetrical barriers of approximately 2.5 and 3.0 kcal/mol. Simulation of 3-cis methyl fentanyl similarly shifted the dihedral landscape in favor of the S2 conformation (S9 Fig). Across our simulations of fentanyl, carfentanil, and lofentanil, all ligands adopted the energetically favored S2 dihedral conformation state within the receptors orthosteric site (Fig 3B).

We next sought to determine the structural basis of fentanyl μOR selectivity over δOR and κOR (Fig 3C) [30]. Only two residues differ between the μOR and δOR orthosteric sites: N129$^{2.63}$ is a lysine and W320$^{7.35}$ is a leucine. W320L$^{7.35}$ removes a large hydrophobic cap above the orthosteric site rendering it more solvent accessible. The substitution N129K$^{2.63}$ introduced a steric clash between the receptor and the n-alkyl phenyl ring of fentanyl such that ligand binding requires the lysine to adopt unfavorable rotameric conformations. Three residues differ between the μOR and κOR orthosteric sites: N129V$^{2.63}$, V302I$^{6.55}$, and W320Y$^{7.35}$. W320Y$^{7.35}$ both introduces steric clash with the ligand and introduces a rigid cap to prevent access to the orthosteric site. Replacement of V302I$^{6.55}$ introduces a slight clash between the amide group of fentanyl, and N129V$^{2.63}$ does not directly interact with the bound ligand. Amongst the fentanyl analogs, lofentanil exhibits the least selectivity while fentanyl and carfentanil are highly selective between the μOR, δOR, and κOR (Fig 3D). These selectivity profiles correlate well with both ligand propensity to adopt the productive S2 dihedral conformation and the degree of steric hindrance to each receptor [30].

## Structure activity relationship

To date, several hundred fentanyl analogs have been synthesized, producing a wealth of historical structure-activity-relationship (SAR) data [29]. We next sought to contextualize this SAR data by docking compounds into a rigid receptor configuration obtained from the simulated fentanyl bound state. The "magnificent four" comprise fentanyl (ED$_{50}$ = 0.011 mg/kg) itself, sufentanil (ED$_{50}$ = 0.00071 mg/kg), alfentanil (ED$_{50}$ = 0.044 mg/kg), and remifentanil (ED$_{50}$ = 0.004 mg/kg) which represent the most common members of the 4-anilidopiperdine series used in the clinical setting for pain management. Sufentanil differs from fentanyl through two modifications to the piperidine scaffold: replacement of the n-alkyl phenyl ring with a thiophene ring and an addition of a 4-methoxymethyl group to the piperidine ring. While the thiopene ring occupies a similar footprint to fentanyl's benzene ring within the hydrophobic cleft of TM2/3, the 4-methoxymethyl serves to both increase the ligand-receptor interaction surface area and decrease the conformational dynamics of the ligand in favor of the productive S2 dihedral conformational similar to that of the 4-carbomethoxy of carfentanil (Fig 4A), consistent with its increased binding potency. In alfentanil, the thiopene ring of sufentanil is replaced with a highly polar 4-ethyl-5-oxo-1H-tetrazol moiety. This n-alkyl tetrazol group is shifted

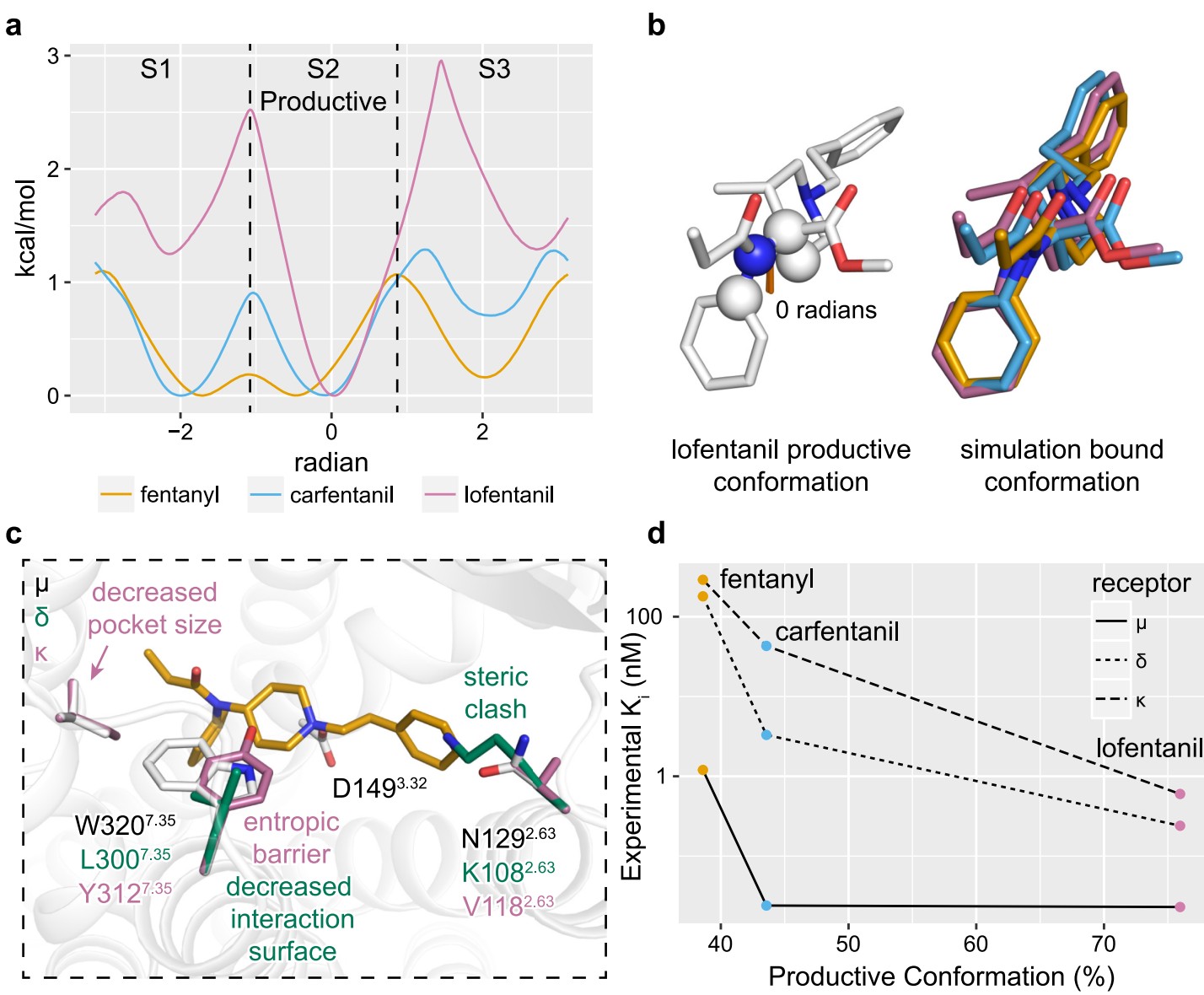

**Fig 3. Piperidine rigidity influences both affinity and selectivity. a,** N-aniline-piperidine connecting dihedral free energy landscapes. S2 dihedral occupancy for fentanyl, carfentanil, and lofentanil was calculated to be 38.5%, 43.4%, and 75.9% respectfully. **b,** Superposition of lofentanil's optimal S2 dihedral conformation with low-energy conformations identified in mABP ligand binding simulations. **c,** Comparison of μOR, δOR, κOR orthosteric site amino acid compositions and selectivity structural analysis. **d,** Correlation between S2 dihedral propensity with *in-vitro* potencies taken from ref [30].

slightly compared to fentanyl and sufentanil's and makes largely unfavorable contacts within the hydrophobic TM2/3 binding cleft (Fig 4A). Remifentanil, like carfentanil ($ED_{50}$ = 0.00032 mg/kg), contains a 4-carbomethoxy-substituted piperidine ring; however the n-alkyl phenyl ring is replaced with a secondary polar carbomethoxy group. The polar n-alkyl carbomethoxy group is buried down towards the receptors core's where the hydroxyl group can form a hydrogen bond Y324[7.43] (Fig 4B).

Addition of a methyl group to the piperidine scaffold yielded one of the more potent fentanyl derivatives, 3-cis-methylfentanyl ($ED_{50}$ = 0.00058 mg/kg) [31]. Within the lofentanil bound pose, the 3-cis methyl group occupies a hydrophobic space between I146[3.29], D149[3.32],

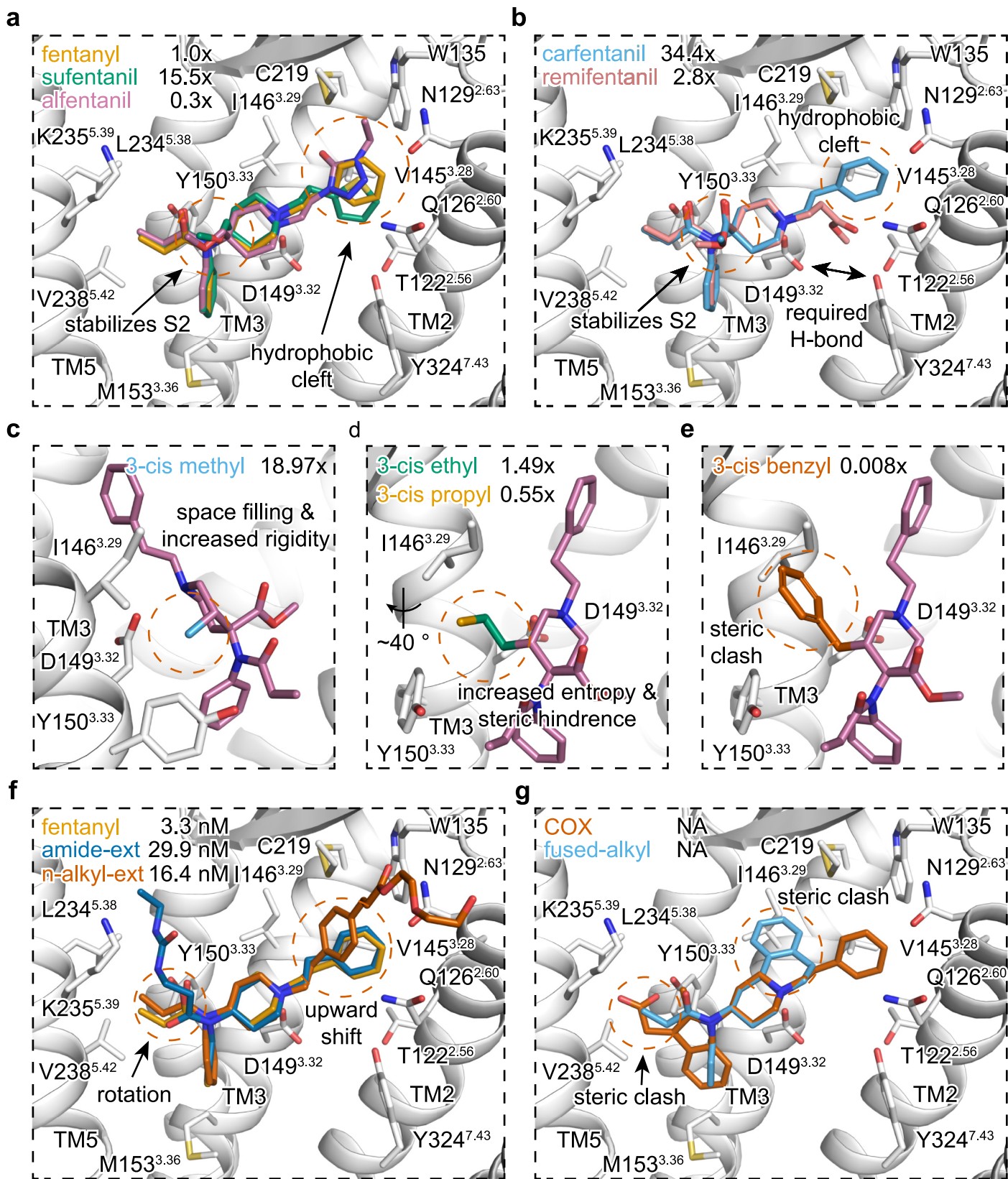

**Fig 4. Docking and modelling of various fentanyl derivatives. a-b,** Docking and sidechain interactions of fentanyl, sufentanil, and alfentanil (a), and carfentanil, and remifentanil (b). Relative potencies (*in-vivo* ED$_{50}$) are taken from ref [29] and are displayed as a fold-difference from fentanyl in the upper left corner of each panel. **c-e,** Modelling of various 3-cis alkyl groups. Relative potencies (*in-vivo* ED$_{50}$) are taken from ref [31] and are displayed as a fold-difference from fentanyl. **f,** Docking and sidechain interactions of fentanyl derivatives bearing amide (Compound 4D [33]) or n-alkyl phenyl (Fen-Acry-PEO$_7$ [32]) extensions. **g,** Modeling of fused-ring fentanyl derivatives.

and Y150$^{3.33}$ and restricts the ligand to the productive S2 dihedral conformation (Fig 4C and S9 Fig). Replacement of 3-cis methyl with larger functional groups such as–ethyl, -propyl, and other alkyl additions introduce increasing sidechain and backbone clash with TM3 as a function of size (Fig 4D and Fig 4E). Unlike 3-cis methyl, 3-trans methyl substation on the piperidine ring (ED$_{50}$ = 0.0094 mg/kg) stabilizes an intermediate ligand conformation found between the S1 and productive S2 dihedral conformations without incurring steric clash with the receptor (S9 Fig). Addition of a 2-cis methyl group (ED$_{50}$ = 0.665 mg/kg) introduces significant steric clash against I146$^{3.29}$ to preclude interaction between the ligand's protonated amine and D149$^{3.32}$.

Recent attempts to synthesize bivalent fentanyl derivatives commonly employ extension or functionalization of the amide or n-alkyl groups at the cost of decreased analgesic activity [29,32,33]. Our bound pose accommodates these large extensions without altering position of the n-aniline or piperidine rings (Fig 4F). Attempts to model conformationally restricted fentanyl derivatives introduced significant steric clash and prevented the ligand from occupying the identified binding pose (Fig 4G), consistent with the fact that these molecules exhibit insignificant analgesic activity [34,35]. Together, our bound pose of fentanyl and its derivatives provide a rational, structure based explanation of SAR for the 4-anilidopiperdine series compounds.

## M153 mediates β-arrestin signaling

Confident in our computationally derived poses of fentanyl, carfentanil, and lofentanil, we next sought to understand the mechanism of ligand induced biased activation of the receptor. Focusing only on the orthosteric pocket, comparison of the DAMGO-bound active µOR-Gi protein complex cryo-EM structure to our model of the receptor bound to the β-arrestin biased agonist fentanyl revealed largely similar side chain arrangements with exception of M153$^{3.36}$ and W295$^{6.48}$ (Fig 5A) [36]. Notably M153$^{3.36}$ is pushed downward by fentanyl's n-aniline ring, but not in the DAMGO structure, to adopt a rotameric conformation which directly displaces W295$^{6.48}$. This fentanyl induced M153$^{3.36}$ rotamer is also not observed in the antagonist bound or morphine derived agonist bound structures of the µOR (S10 Fig) [27,28]. Thus, we hypothesized that fentanyl's unique displacement of M153$^{3.36}$ may mediate its β-arrestin biased signaling.

To investigate the role of M153$^{3.36}$ in β-arrestin biased signaling, we constructed a series of hydrophobic mutations at the M153$^{3.36}$ position of decreasing sidechain size and assessed ligand-induced assembly of the receptor with Gi protein complex or with arrestin using a direct interaction NanoBiT assay [37]. All five mutations substantially reduced the EC$_{50}$ of Gi coupling for both DAMGO and fentanyl, which displayed a correlation with the size of side chain of the mutated residue (Fig 5B). Interestingly, while fentanyl requires M153$^{3.36}$ to achieve Gi coupling above 55%, DAMGO was less sensitive to side-chain replacement and retained near 100% coupling for larger amino acid side chains M153F/L/I$^{3.36}$.

Unexpectedly, β-arrestin coupling for both fentanyl and DAMGO exhibit a shared dependence upon M153$^{3.36}$. For the mutations assessed, all but larger amino acid side chains M153F$^{3.36}$ and M153F/L$^{3.36}$ for DAMGO and fentanyl, respectfully, failed to elicit coupling.

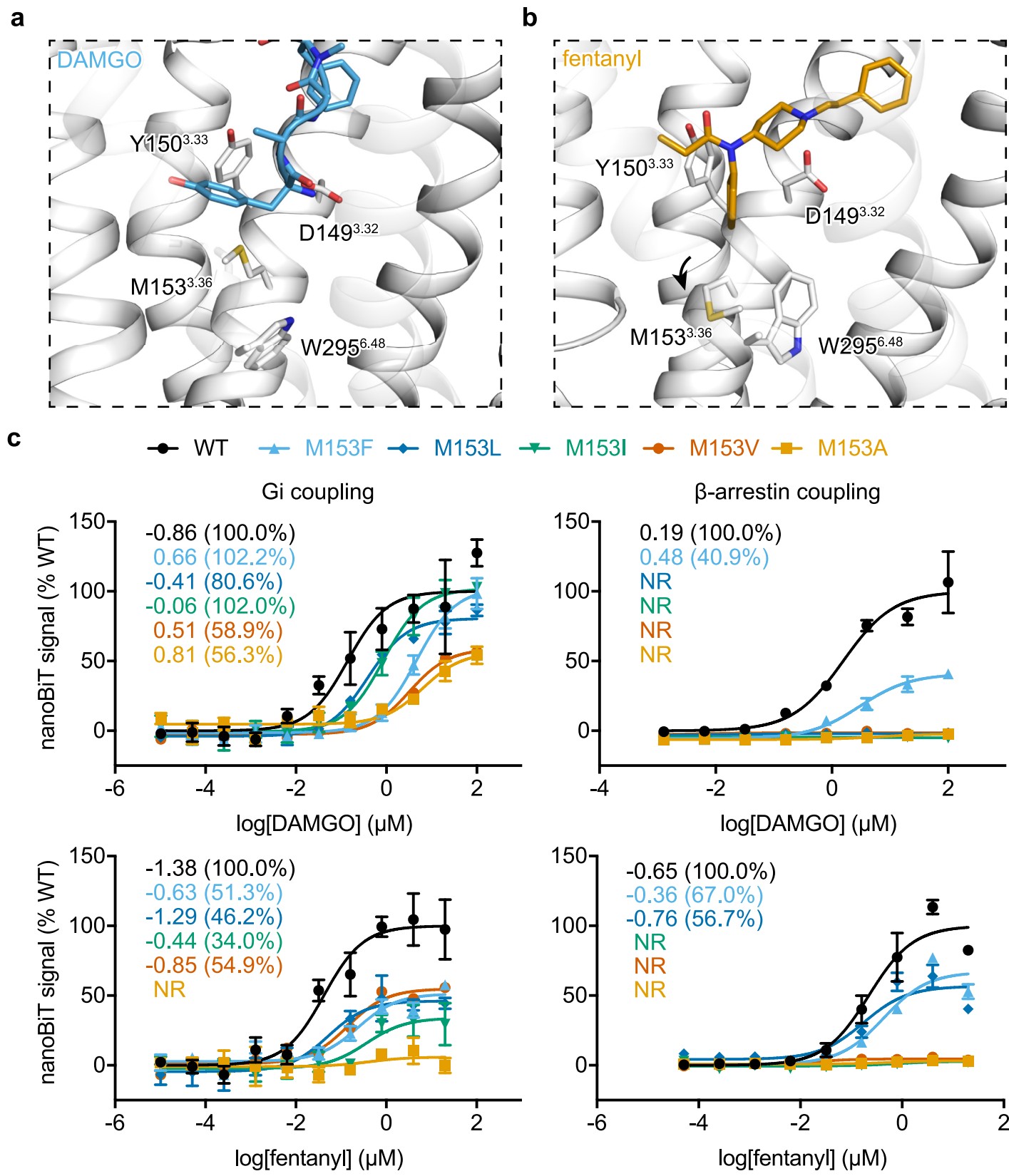

**Fig 5. Effects of M153 mutations on ligand induced β-arrestin and Gi complex coupling. a,** Side-by-side structural comparison of orthosteric site interactions between DAMGO (PDB code: 6DDE) and fentanyl bound µOR. **b-c,** Dose-response curve for DAMGO and fentanyl induced Gi complex and β-arrestin coupling for wildtype and M153$^{3.36}$ mutant receptors measured by nanoBiT direct interaction assays. EC$_{50}$ and maximal coupling efficacy compared to DAMGO are shown. Values and error bars reflect mean ± s.e.m. normalized to DAMGO of three technical replicates.

and those that retained coupling saw severely reduced maximal levels (Fig 5B). Together, these results suggest that M153$^{3.36}$ is required for β-arrestin, but not Gi coupling.

### Fentanyl's n-aniline ring mediates β-arrestin signaling

Given the fact that fentanyl's n-aniline ring is directly packed against M153$^{3.36}$, a key residue for β-arrestin signaling, we next sought to determine if fentanyl's n-aniline ring is required for β-arrestin recruitment (Fig 6A). A series of fentanyl derivatives (FD-1, -2, -3) were synthesized where the n-aniline ring was replaced with smaller aliphatic functional groups. This series of compounds were predicted to have reduced packing effects on the rotameric conformation of M153$^{3.36}$ (Fig 6B).

All three derivatives exhibited significantly reduced EC$_{50}$ and maximal Gi coupling compared to fentanyl, where ligand potency increased as a function of aliphatic group size (Fig 6C). One compound in particular, propyl substituted FD-3, retained partial-agonist properties comprising a mid-nanomolar EC$_{50}$ with a maximal coupling efficiency that is approximately 67% that of morphine [16] and comparable to the M153I$^{3.36}$ mutant. All of our derivatives were completely Gi biased and failed to elicit any detectable level of β-arrestin coupling, up to 4 µM (Fig 6C). Taken together with our prior mutational analysis, these experiments suggest that the n-aniline ring is required for fentanyl mediated β-arrestin, but not Gi coupling, and that this effect is mediated through an interaction with M153$^{3.36}$.

## Discussion

Enhanced sampling methods for molecular dynamics (MD) are often invoked to accelerate simulations of rare events for biological systems [38], such as ligand binding and release. In this study, we implemented a previously derived efficient mABP scheme [17] with minimal communication to the GPU enabled AMBER MD engine, which significantly reducing the computational time required to sample rare events while providing high-density simulation throughput. In total, we performed 92 µs of mABP simulations on our cluster comprising two nodes with 8 Nvidia GTX 1080 TI GPUs (S1 Table). Running at 110–120 ns/day for the GPCR-ligand binding systems, each simulation required approximately 16–17 days per simulation per GPU to complete.

During our initial benchmarking and system testing simulations of carazolol binding to β2AR, we found that overfill protection [20] was essential to simulation success. Overfill protection prevents trajectory "spoiling", arising from the bias potential pushing the ligand through the protein causing irreversible structural deformation events. Interestingly, a relationship between bias potential fill-limit and successful ligand binding events was observed. Simulations performed at lower fill-limit resulted in no binding events, whereas those performed at higher fill-limit resulted in many ligand entry events into the orthosteric pocket, but failed to identify any dominant bound conformation. For µOR binding simulations, we arrived at a fill-limit lower than that used for β2AR simulations. Unlike the orthosteric site of β2AR which is partially occluded by the receptor's extracellular loop 2, the µOR has no such barrier above the orthosteric site. These experiments suggest a careful fill-limit calibration is required to ensure that the bias potential is high enough to sample rare events without spoiling the trajectory. Likely, this value varies between GPCRs and potentially between different ligands.

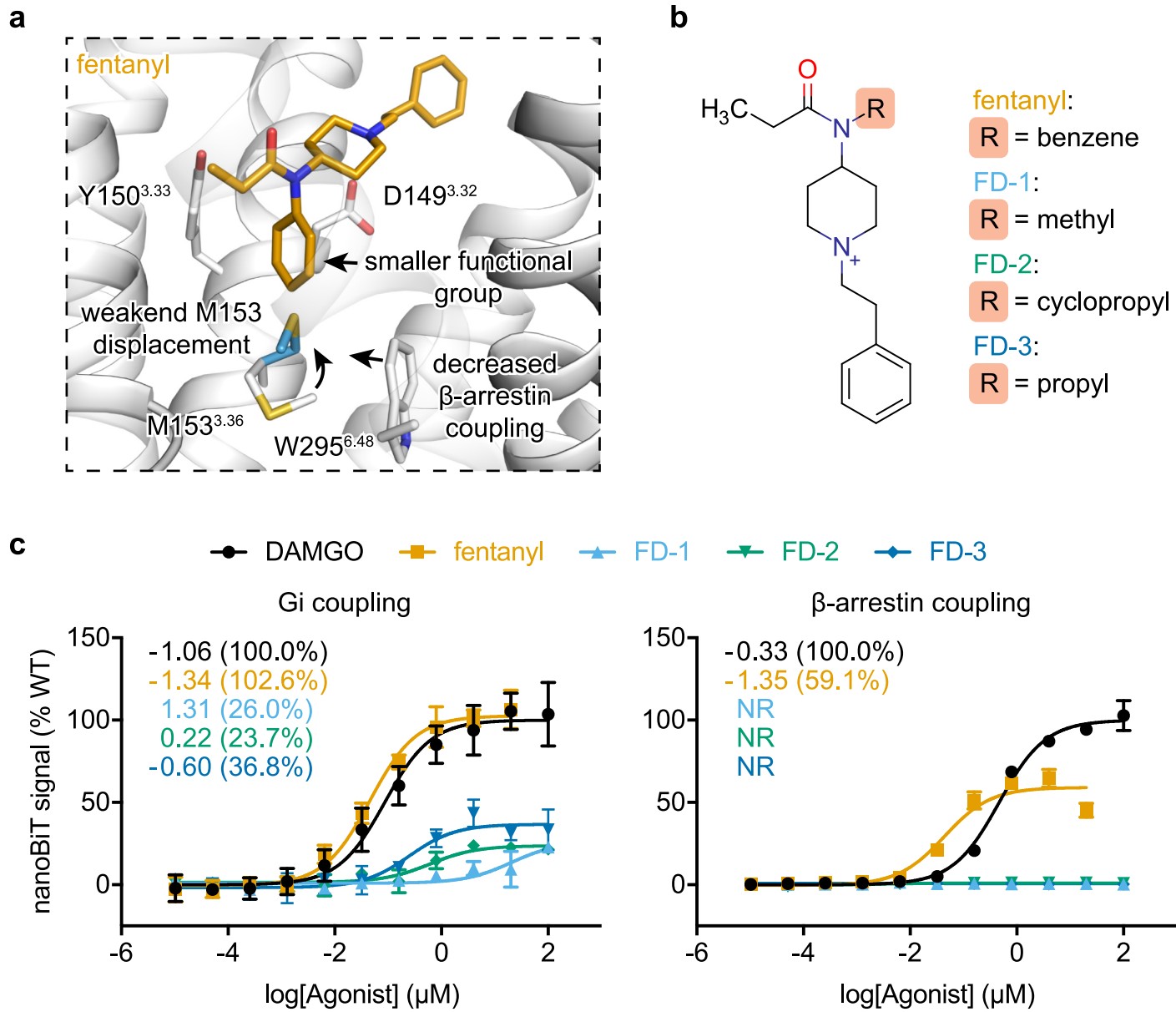

**Fig 6. Fentanyl derivatives with no β-arrestin signaling. a-b,** Structure based rational and chemical structure of newly synthesized fentanyl derivatives. **c,** Gi complex and β-arrestin coupling dose-response curve for DAMGO, fentanyl, and compounds FD1-3 measured by nanoBiT direction interaction assays. $EC_{50}$ and maximal coupling efficacy compared to DAMGO are shown. Values and error bars reflect mean ± s.e.m. normalized to DAMGO of three technical replicates.

Simulations of fentanyl, carfentanil, and lofentanil identified a common shared pose within the receptor's orthosteric site that makes chemically favorable interactions with the μOR (Fig 2B and Fig 2C). As the ligand's scaffold comprising the n-aniline, piperidine, and n-alkyl phenyl rings remains unchanged, the degree of within- and between-ligand convergence across independent mABP simulations largely improves the statistical confidence in the bound conformation. Importantly, each ligand's protonated amine was found to form a salt-bridge with D149[3.32]. As with other aminergic GPCRs, interaction with this fully conserved D[3.32] provides a critical anchoring point for ligand orientation and is an essential feature for receptor activation [18]. We note that recent attempts to "dock" fentanyl to the μOR crystal structure and

long-timescale simulations of TRV-130, a recently described μOR agonist totaling over 44 μs of simulation, did not observed this key interaction [39,40]. For studies that correctly identify the $D^{3.32}$ salt bridge, the resulting pose is sterically incompatible with fentanyl derivatives bearing n-alkyl or amide extensions [41,42]. The mABP approach employed here offers a major advantage over virtual screening for targeted compound screening by employing a fully dynamic, unrestrained, explicitly solvated membrane environment which is coupled with the statistical power or mABP for improved confidence in identified bound poses [43]. For these reasons, this technique has also proved valuable for systems with induced-fit binding mechanics [44].

With any method requiring construction of collective variables, one accepts that sampling efficiency is dependent on making a good construction. In the present case, the collective variable has a symmetry that can be realized by exchanging identities ("blue" and "red" in S1 Fig) of the two RMSD reference points. Our particular construction appears to be the best one, evidenced by the significant agreement among our many repeated simulations and the fact that the blue CV atoms in S3 Fig ended up nearest the blue reference point. The 50/50 odds played out in our favor here, but in general one could explore both CV constructions with trial simulations.

A poorly constructed CV decreases the efficiency of the mABP method but the proportionality of this decrease is completely unknown and entirely dependent on the application. We report in SI that switching the references in our simulations does reduce efficiency (S11 Fig). Unfortunately, it is impossible to quantify the efficiency loss. Through comparison to data generated by large-scale brute force simulation using Anton [25], we estimate our mABP approach with good CV to be anywhere from 8 to 18 times more efficient when comparing simulations of ligand binding to β2AR (S2 Table). Without running the alternate CV for at least 8 times longer than our production trajectories, we cannot assess the efficiency. The reversed CV are less efficient, but it is infeasible to measure this efficiency.

Beyond their described roles of increasing the ligand-receptor interaction surface area, we also sought to understand how modifications to the piperidine ring affect conformational heterogeneity around the n-aniline and piperidine rings. All tested piperidine additions reduced conformational heterogeneity, with the 3-cis methyl group contributing a majority of this effect over 4-carboxymethyl. As lofentanil exhibited only one energetically favorable dihedral conformation, we hypothesize the S2 conformation is productive for both ligand binding and receptor activation (Fig 3A and Fig 3B). Indeed, all of our μOR bound poses for fentanyl, carfentanil and lofentanil adopted the S2 dihedral state within the receptors orthosteric site. These simulations not only inform on the structural influence of piperidine scaffold modifications, but also provide strong support in favor of the common binding conformation for fentanyl and its derivatives in the orthosteric site.

Fentanyl and its derivatives display robust affinity and selectivity for the μOR over the δOR and κOR despite limited perturbations to the orthosteric site amino acid composition [30]. By modelling the δOR orthosteric site onto our fentanyl bound pose, we proposed $N129K^{2.63}$ to be the largest contributor to the decreasing binding affinity. The larger side chain of the lysine perturbs the shape of the orthosteric binding pocket. Unlike the δOR, the only possible $W320Y^{7.35}$ rotamer in κOR introduces direct steric clash with the ligand's rigid core. Based on this structural analysis, an ordered ranking affinity for fentanyl derivatives was predicted as μOR > δOR > κOR, consistent with observed *in-vivo* potencies [30].

We next sought to provide a rational structure-based explanation for historical SAR of the fentanyl scaffold (Fig 4). Modifications to the 4-position on the piperidine ring were found to increase binding affinity by both increasing the ligand-receptor interface and propensity to adopt the productive S2 dihedral conformation, consistent with the increased efficacy seen in

sufentanil over fentanyl. This increased potency can be offset by replacing the n-alkyl group with a polar moiety, which introduces repulsive interactions to the hydrophobic TM2/3 cleft. This change accounts for the affinity decrease from sufentanil to alfentanil and from carfentanil to remifentanil. These polar groups may also interfere with formation of a hydrogen bond between D149[3.32] and Y324[7.43] observed in both crystal and cryo-EM structures of agonist bound μOR [28,36]. As agonists stabilize this conserved hydrogen bond [18,45], we speculate that its disruption by polar n-alkyl groups will negatively affect receptor activation. In addition, fentanyl's bound conformation provides a rational explanation for alkyl substations around the piperidine scaffold. We further explored compatibility of fentanyl analogs with amide and n-alkyl extensions. While these functionalizations greatly diminished μOR activity [32,33], we successfully docked examples of both ligand types in conformations that do not alter the core-ligand interaction interface comprising the n-aniline and piperidine rings. As we previously noted, prior studies have been unable to identify bound conformation- compatible amide and n-alkyl extensions [41,42]. Lastly, we found that rigid n-aniline and fused-alkyl fentanyl derivatives prevent ligand binding through the introduction of severe steric clash, consistent with their inability to induce antinociceptive properties [34,35]. Together, our bound pose of fentanyl and its derivatives provide a rational, structure-based explanation of SAR for the 4-anilido-piperdine series compounds which further support the bound conformation of fentanyl as observed in our simulations.

Comparison of fentanyl and its derivatives to structures of agonist- and antagonist-bound μORs revealed a unique M153[3.36] rotameric conformation previously unobserved (Fig 5) [27,28,36]. Mediated by the common n-aniline ring, the rotamers of M153[3.36] directly influence the conserved W295[7.35] microswitch, critical for class A GPCR receptor activation. Mutational studies revealed a unique sensitivity to M153[3.36] perturbations for fentanyl-mediated but not DAMGO-mediated, maximal Gi coupling levels. As fentanyl's n-aniline ring is buried deeply within the orthosteric site to displace M153[3.36] unlike DAMGO, this sensitivity is consistent with each ligand's spatially distinct binding footprint. We thus hypothesized that fentanyl's ability to displace M153[3.36] is required for β-arrestin, but not Gi, coupling (Fig 6). We reasoned that replacing the bulky n-aniline ring of fentanyl with smaller aliphatic groups would diminish the ligand's β-arrestin biased signaling profile. Indeed, all of the fentanyl analogs failed to induce β-arrestin coupling while one compound in particular, propyl-substituted FD-3, retained partial Gi coupling levels with nanomolar efficacy, comparable to M153[3.36] mutations. We propose a mechanism where the n-aniline ring of fentanyl mediates μOR β-arrestin through a novel M153[3.36] microswitch. Together, our study provides structural insight into fentanyl-mediated β-arrestin biased signaling and provides chemical insights to the design of novel fentanyl-based analgesics that exhibit complete bias toward Gi coupling.

In summary, through adaptively biased molecular dynamics simulations we have derived a common binding pose for the fentanyl derivatives within the μOR orthosteric ligand binding pocket. This binding pose is consistent with the vast SAR data from large numbers of fentanyl-related compounds. Moreover, this fentanyl binding mode has allowed us to determine the critical role of a single residue, M153[3.36], for μOR β-arrestin biased signaling, which was further validated by fentanyl derivatives that are capable of mediating G protein signaling but specifically devoid of β-arrestin biased signaling. Beyond application to the μOR–fentanyl system, our findings also have broader impacts for our understanding of biased signaling in other class A GPCRs. Similarly, subtle changes in the chemical structures of β2AR ligands [46] or single residue mutation in the serotonin receptor 5-HT_{2B} [47] can also alter the biased signaling specificity, demonstrating that GPCR signaling specificity can be altered by very minor changes either in the receptor itself or in the chemical structures of the bound ligands.

## Methods

### fABAMBER benchmarks

All-atom atmospheric simulations of fentanyl solvated in a TIP3P water box comprising 150 mM NaCl of various atom counts were performed in the NPT ensemble for 500 ns with periodic boundary conditions and the CHARMM36m forcefield [48]. Comparative simulations were performed with AMBER's native pmemd.cuda, fABAMBER's pmemd.cuda, and AMBER with NFE enabled. fABAMBER and NFE enabled simulations employed an RMSD collective variable where the ligand was decomposed into two CVs (S3 Fig) where each atoms reference was located on its initial configuration within the simulation system. For mABP simulations with fABAMBER, the CV space was discretized into a 480 x 480 grid and ranged from 0 to 60 Å for a bin width of 0.125 Å. Biasing parameters were b = 0.9, c = 0.005/ $\delta t$, $\alpha$ = 20. For NFE simulations, the default biasing settings for 'FLOODING' were used as shown in figure 22.9 of the NFE manual (http://ambermd.org/doc12/nfe.pdf) were used. For all simulations, a 2 fs time-step was used and bonds involving hydrogen atoms were constrained by SHAKE with non-bonded interactions cut at 8 Å. Trajectory frames saved every 10 ps. Temperature and pressure were maintained at 310 K and 1 bar by the Langevin thermostat and Berendsen barostat with isotropic coupling.

### Molecular dynamics simulation setup and equilibration

All-atom atmospheric simulations of the human β2AR (PDB Code: 2RH1) and μOR (PDB Code: 4DKL) were performed in the NPT ensemble with periodic boundary conditions and the CHARMM36m forcefield [48]. From the inactive crystal structures, both receptors were prepared by removing all non-GPCR protein chains, fusions partners, and heteroatoms including the crystallographic ligand. Mouse μOR was humanized and subjected to 5,000 rounds of model generation using Modeller9.18 [49]. The crystallographically unresolved intracellular loop 3 (ICL3) was left unbuilt for both receptors. Protonation states of titratable residues were set in agreement with prior simulations of inactive GPCRs [50] and histidines were modelled with an explicit proton on their epsilon nitrogen. The allosteric sodium interacting with $D^{2.50}$ was modeled to further mimic the receptors inactive conformation [51]. Each receptor was then capped with neutral acetyl and methylamine groups and embedded into a pre-equilbirated palmitoyl-oleoyl-phosphatidylcholine (POPC) lipid bilayer, solvated in a box of TIP3P waters allowing for 14 Å of padding on all sides with 150 mM NaCl, and neutrailized by removing appropriate ions or counter ions using the Desmond system builder within Maestro (Schrödinger Release 2018–1: Maestro, Schrödinger, LLC, New York, 2018).

Prior to mABP production simulations, 25,000 steps of energy minimization were carried out followed by equilibration in the NVT and NPT ensembles for 10 and 50 ns respectively with harmonic restraints (10 kcal mol$^{-1}$ Å$^{-2}$) placed on all Cα atoms. Temperature and pressure were maintained at 310 K and 1 bar by the Langevin thermostat and Berendsen barostat with semi-isotropic coupling. An additional 50 ns of unrestrained NPT simulation was performed with the Monte Carlo barostat. For all simulations, a 2 fs time-step was used and bonds involving hydrogen atoms were constrained by SHAKE with non-bonded interactions cut at 8 Å. Trajectory frames saved every 10 ps. After equilibration, ligands were placed approximately 10 Å above the orthosteric site. Ligand parameters were generated by the CHARMM General Force Field program [52].

### Ligand binding simulations and pose extraction

To accelerate sampling of ligand binding, we implemented our previously described mABP scheme [17] with overfill protection [20] into the GPU-enabled pmemd.cuda molecular

dynamics [21] engine for both AMBER16 [22]. Two RMSD CVs were defined between each ligand and the receptor (S1 Fig and S3 Fig). The receptor reference points of CV1 and CV2 are two dynamically updating single centers-of-geometry comprising Cα atoms found below the orthosteric site in TM2/3/7 and TM3/5/6 respectfully. From the geometric center of the CV1 and CV2 references, a receptor center is dynamically calculated and serves as the origin for a cylindrical restraint (radius = 18 Å). This restraint is combined with a maximal allowable CV value of 22 Å to prevent unwanted ligand diffusion away from the receptor and into periodic images. The collective variable space was discretized into a 480 x 480 grid and ranged from 0 to 60 Å for a bin width of 0.125 Å. Biasing parameters were b = 0.9, c = 0.005/ $\delta t$, $\alpha$ = 20. mABP production simulations of in replicates of 2 μs were performed in the NTP ensemble. All simulation parameters and trajectories are available upon request. fABAMBER patches for AMBER16 are can be found on Github (https://github.com/ParkerdeWaal/fABAMBER).

Representative pose extraction for each ligand was performed as follows: The free-energy landscape of simulations that identified energetic minima within the orthosteric site were combined and averaged. Simulations without ligand binding events were discarded. A square region comprising the averaged minima was defined and all trajectory frames within this CV space were extracted into a combined subtrajectory comprising only the ligand. The most representative ligand configuration was determined using the DBSCAN clustering algorithm [53] within CPPTRAJ [54] where each cluster required a minimum of 25 points with a distance cutoff between points of 0.7 Å.

### Fentanyl dihedral simulations

Fentanyl and its derivatives were individually solvated in a box of TIP3P waters with a minimum of 8 Å padding and neutralized with the addition of a single sodium ion. Each system was equilibrated in the NVT and NPT ensembles for 10 and 10 ns respectfully using largely the same temperature and pressure coupling schemes used for GPCR simulations where only the pressure-coupling scheme was changed to isotropic. A single common dihedral CV for all ligands was defined by atoms connecting the n-aniline and piperidine scaffold and discretized into a 300 x 300 grid ranging from 0 to $2\pi$ radians. Biasing parameters were b = 0.8, c = 0.1/ $\delta t$, $\alpha$ = 5. mABP simulations lasting 500 ns were performed in the NTP ensemble for each ligand.

### Fentanyl derivative docking

Flexible ligand docking of fentanyl derivatives into a rigid fentanyl bound μOR receptor snapshot was performed using DOCK6.9 [55]. Prior to docking, the W320[7.35] rotamer was adjusted to accommodate 4-position substations to the piperidine scaffold, each ligand was energy minimized, and partial charges were derived using CHIMERA [56].

### Plasmid construction

The human μOR, Gαi1, Gβ1, Gγ2, and β-arrestin2 cDNAs were codon-optimized and synthesized by GENEWIZ (Suzhou, China). The site-directed mutations were introduced into μOR (M153A, M153F, M153I, M153L, and M153V) using PCR.

For β-arrestin recruitment assay, the cDNAs of human μOR (wild type and M153 mutants), as well as pre-activated β-arrestin2 harboring 3A mutations (residues 1–393, I386A, V387A and F388A) (Kang et al., 2015) were inserted into a NanoBiT PPI plasmid (Promega). The large subunit (LgBiT) was positioned at the C-termini of receptor, and the small subunit (SmBiT) was positioned at the N-terminus of β-arrestin2 (3A).

For Gi activation assay, the cDNAs of receptors were subcloned into a modified pfastbac1 vector (Invitrogen), which contained an expression cassette with a haemagglutinin (HA) signal

sequence at the N terminus and LgBiT at the C terminus. The human cDNA of Gαi1, Gβ1 and Gγ2 were separately inserted into pfastbac1 vector, while the SmBiT was fused to C terminus of Gβ1 subunit.

## Sf9 cell membrane preparation and Gi activation assay

High-titer recombinant baculovirus ($>10^9$ virus particles per ml) of μOR (wild type and mutants), Gαi1, Gβ1 and Gγ2 was obtained using Bac-to-Bac Baculovirus Expression System (Invitrogen) as previously described [57,58]. Cell suspensions were cultured for 4 days while shaking at 27°C to generate P1 virus. Sf9 cells at a density of $2\times10^6$ cells/ml were co-infected with P2 virus stock of receptor, Gαi1, Gβ1 and Gγ2 at a volume ratio of 1:1:1:1. The cells were harvested by centrifugation after 48 hours post-infection.

Sf9 cell membranes were disrupted by homogenization in a hypotonic buffer containing 10 mM HEPES (pH 7.5), 10 mM $MgCl_2$, 20 mM KCl and protease inhibitor cocktail (Vazyme, Nanjing, China). Cell membranes were then separated by ultracentrifugation at 45,000 g/min for 30 min. The membrane pellets were resuspended in Phosphate Buffer Saline ($1\times$PBS, pH 7.4) and stored at -80°C until use.

A modified NanoBiT assay was performed to evaluate Gi activation [37]. Briefly, sf9 membrane containing μOR or its mutants were added into 384-well white plates (Greiner) at 20 μl/well. NanoLuc substrates (Promega, after 1:20 dilution) were added at 10 μl/well followed by incubation by 10 μl of 4×compound solutions for 15 min at room temperature. The luminescence was measured at 700 nm using a Perkin-Elmer EnVision plater reader. Results were plotted and analyzed in GraphPad Prism 7.0.

## AD-293 cell transfection and β-arrestin recruitment assay

The human embryonic kidney AD-293 cells (Agilent) were cultured in growth medium that consisted of Dulbecco's modified Eagle's medium (DMEM) with 10% fetal bovine serum (Invitrogen) at 37°C in the presence of 95% $O_2$ and 5% $CO_2$. Cells were plated at a density of $5\times10^5$ cells/well of a 6-well plate for overnight. Using FuGENE HD (Promega), cells were transfected with 1.5 μg receptor plasmid and 1.5 μg β-arrestin2 (3A) plasmid per well and then cultured for 24 hours.

A modified NanoBiT assay was performed to evaluate β-arrestin recruitment [37]. On the day of assay, transfected cells were harvested and resuspended using Opti-MEM (Invitrogen). The cell density was adjusted to $4\times10^5$ cells/ml before plating cells into 384-well white plates (Greiner) per 20 μl/well. 10 μl/well of NanoLuc substrate (Promega, after 1:20 dilution) was added followed by addition of 10 μl/well indicated concentration of 4×compounds solutions. After 15 min incubation, luminescence was recorded in Perkin-Elmer EnVision plate reader, and curves were generated and analyzed using Graphpad Prism 7.0.

## Fentanyl derivative synthesis

**Methods for synthesis of compound FD-1.** The appropriate anhydride (1.2 equiv) was added slowly to a stirred 0.5 M solution of compound **1a** (1 mmol) and Et$_3$N (2.0 equiv) in $CH_2Cl_2$ at RT. After having been stirred for 5 h, the reaction mixture was quenched with water. The organic layer was washed with saturated $NaHCO_3$ (aq), 1 M HCl, and brine. The extract was dried over $Na_2SO_4$ and concentrated to provide amides **2a** which was used directly.MS (m/e): 271.3 (M+1); Trifluoroacetic acid (1 mL) was added dropwise to a solution of **2a** (1 mmol) in 1 ML of dry $CH_2Cl_2$ at 0°C. Then the rmixture was stirred at RT for 3h. After removal of the solvents, the residue was dissolved in $CH_3CN$ (5 mL), $K_2CO_3$ (5 eq) and (2-bromoethyl)benzene (1.2eq) was added. The mixture was stirred at RT for 10 h. Filtered

and the filtrate was concentrated. The crude product was purified by preparative thin layer chromatography ($CH_2Cl_2$/MeOH, 20:1) to afford **FD-1** (0.19g, 69%). MS (m/e): 275.5 (M+1); $^1$H NMR (400 MHz, CDCl3) δ: 7.32–7.14 (m, 5H), 4.60–4.45 (m, 0.7H), 3.64–3.52 (m,0.3H), 3.14–2.98 (m, 2H), 2.89–2.72 (m, 5H), 2.62–2.53 (m, 2H), 2.41–2.23 (m,2H), 1.97–1.53 (m,4H), 1.1 5(q, 3H).

**Methods for synthesis of compound FD-2.**    The appropriate anhydride (1.2 equiv) was added slowly to a stirred 0.5 M solution of compound **1b** (1 mmol) and $Et_3N$ (2.0 equiv) in $CH_2Cl_2$ at RT. After having been stirred for 5 h, the reaction mixture was quenched with water. The organic layer was washed with saturated $NaHCO_3$ (aq), 1 M HCl, and brine. The extract was dried over $Na_2SO_4$ and concentrated to provide amides **2b** which was used directly.MS (m/e): 299.4 (M+1); Trifluoroacetic acid (1 mL) was added dropwise to a solution of **2b** (1 mmol) in 1 ML of dry $CH_2Cl_2$ at 0˚C. Then the rmixture was stirred at RT for 3h. After removal of the solvents, the residue was dissolved in $CH_3CN$ (5 mL), $K_2CO_3$ (5 eq) and (2-bromoethyl)benzene (1.2eq) was added. The mixture was stirred at RT for 10 h. Filtered and the filtrate was concentrated. The crude product was purified by preparative thin layer chromatography ($CH_2Cl_2$/MeOH, 20:1) to afford **FD-2** (0.21g, 70%). MS (m/e): 303.4 (M+1); $^1$H NMR (400 MHz, CDCl3) δ: 7.35–7.16 (m, 5H), 4.54–4.41 (m, 0.7H), 3.64–3.50 (m,0.3H), 3.23–3.01 (m, 4H), 2.87–2.75 (m, 2H), 2.66–2.50 (m, 2H), 2.42–2.30 (m,2H), 2.20–2.01 (m,2H), 1.95–1.49 (m, 6H), 1.21–1.13 (m, 3H), 0.95–0.85 (m, 3H).

**Methods for synthesis of compound FD-3.**    The appropriate anhydride (1.2 equiv) was added slowly to a stirred 0.5 M solution of compound **1c** (1 mmol) and $Et_3N$ (2.0 equiv) in $CH_2Cl_2$ at RT. After having been stirred for 5 h, the reaction mixture was quenched with water. The organic layer was washed with saturated $NaHCO_3$ (aq), 1 M HCl, and brine. The extract was dried over $Na_2SO_4$ and concentrated to provide amides **2c** which was used directly.MS (m/e): 297.3 (M+1); Trifluoroacetic acid (1 mL) was added dropwise to a solution of **2c** (1 mmol) in 1 ML of dry $CH_2Cl_2$ at 0˚C. Then the rmixture was stirred at RT for 3h. After removal of the solvents, the residue was dissolved in $CH_3CN$ (5 mL), $K_2CO_3$ (5 eq) and (2-bromoethyl)benzene (1.2eq) was added. The mixture was stirred at RT for 10 h. Filtered and the filtrate was concentrated. The crude product was purified by preparative thin layer chromatography ($CH_2Cl_2$/MeOH, 20:1) to afford **FD-3** (0.20g, 67%). MS (m/e): 301.4 (M+1); $^1$H NMR (400 MHz, CDCl3) δ: 7.34–7.18 (m,5H), 4.26–4.04 (m, 1H), 3.14–3.02 (m,2H), 2.87–2.75 (m, 2H), 2.66–2.40 (m, 5H), 2.19–1.94 (m, 4H), 1.85–1.70 (m,2H), 1.15 (t, 3H), 0.94–0.8 (m, 4H).

## Supporting information

**S1 Fig. Detailed mABP simulation setup and binding of carazolol to β2AR. a-b,** graphical representation of collective variables and cylindrical restraints used in mABP simulations. **c,** Averaged free energy landscape for carazolol binding to the β2AR. Contours are drawn at 1, 3, and 5 kcal/mol. **d,** Structural overlap of carazolol's predicted low-energy conformation (white cartoon; orange sticks) and crystal structure (PDB Code: 2RH1; light blue sticks). (PDF)

**S2 Fig. Carazolol fill limit calibration. a,** Distance between carazolol's protonated amine to D113$^{3.32}$ for various bias potential fill-limits. **b,** Free energy landscapes of mABP simulations where the bias potential was capped at 17 kcal/mol. Contours are drawn at 1, 3, and 5 kcal/ mol. (PDF)

**S3 Fig. BU72 and fentanyl collective variables. a-b,** Ligand atom selections for CV1 and CV2 for BU72 and fentanyl.
(PDF)

**S4 Fig. BU72 mABP simulations. a,** Free energy landscapes of simulation replicates. Contours are drawn at 1, 3, and 5 kcal/mol. **b,** Distance between BU72's protonated amine to D149$^{3.32}$.
(PDF)

**S5 Fig. Fentanyl mABP simulations. a,** Free energy landscapes of simulation replicates. Contours are drawn at 1, 3, and 5 kcal/mol. **b,** Distance between fentanyl's protonated amine to D149$^{3.32}$.
(PDF)

**S6 Fig. Carfentanil mABP simulations. a,** Free energy landscapes of simulation replicates. Contours are drawn at 1, 3, and 5 kcal/mol. **b,** Distance between carfentanil's protonated amine to D149$^{3.32}$.
(PDF)

**S7 Fig. Lofentanil mABP simulations. a,** Free energy landscapes of simulation replicates. Contours are drawn at 1, 3, and 5 kcal/mol. **b,** Distance between lofentanil's protonated amine to D149$^{3.32}$.
(PDF)

**S8 Fig. Structural comparison of BU72 and carfentanil. a,** Side-by-side structural and pharmacophore comparison between BU72 (PDB code: 5C1M) and carfentanil bound μOR.
(PDF)

**S9 Fig. Extended n-aniline-piperidine ring dihedral free energy landscape. a,** S2 dihedral occupancy for fentanyl, carfentanil, and lofentanil, 3-cis methyl fentanyl, and 3-trans methyl fentanyl was calculated to be 38.5%, 43.4%, 75.9%, 56.3%, and 34.7% respectfully.
(PDF)

**S10 Fig. Orthosteric site structural comparison of antagonist, agonist, and fentanyl bound structures. a,** Side-by-side comparison of antagonist (PDB code: 4DKL), agonist (PDB code: 5C1M), and predicted fentanyl bound pose. M153$^{3.36}$ and W295$^{6.48}$ side-chain rearrangements are annotated.
(PDF)

**S11 Fig. BU72 and carazolol ligand binding simulations with switched CV reference points. a,** Distance between carazolol's protonated amine to D113$^{3.32}$. **b,** Distance between BU72's protonated amine to D149$^{3.32}$. **c,** Free energy landscapes of for BU72 simulations with switched CV reference points. Contours are drawn at 1, 3, and 5 kcal/mol. **d,** Structural comparison of the most populated conformation of simulation 2 compared to the crystal structure of BU72.
(PDF)

**S12 Fig. Reaction schemes for FD-1, -2, and -3.**
(PDF)

**S1 Table. List of simulations.** * Indicates simulations where CVs are switched.
(DOCX)

**S2 Table. Relative sampling efficiency of biased and unbiased ligand binding simulations.** Binding events, including both ligand binding and unbinding from the receptor, were determined by an interatom distance between the carboxylic acid carbon on Asp$^{3.32}$ and the protonated amine found on each ligand. To be considered bound the inter-atom distance must be less than 4 Å, and to be considered unbound the inter-atom distance must be greater than 10 Å. For carazolol, we computed efficiency based on simulations 1–4 (flim 17 kcal/mol). For BU72, we computed efficiency based on simulations 1–4. For Dror et al., binding events included poses 4, 4', 4", and 5 as defined in [25].
(DOCX)

**S1 File. PDB of fentanyl bound pose.**
(PDB)

**S2 File. PDB of carfentanil bound pose.**
(PDB)

**S3 File. PDB of lofentanil bound pose.**
(PDB)

## Acknowledgments

This work was inspired by National Public Radio's, and Michigan Public Radio's (WVGR 104.1 FM), continuous coverage of the national opioid overdose epidemic. Computation for the work described in this paper was supported by the High Performance Cluster and Cloud Computing (HPC3) Resource at the Van Andel Research Institute.

## Author Contributions

**Conceptualization:** Parker W. de Waal, H. Eric Xu, Bradley M. Dickson.

**Data curation:** Parker W. de Waal.

**Formal analysis:** Parker W. de Waal, Bradley M. Dickson.

**Investigation:** Parker W. de Waal, Jingjing Shi, Erli You, Xiaoxi Wang, Yi Jiang, Bradley M. Dickson.

**Methodology:** Parker W. de Waal, Jingjing Shi, Yi Jiang, H. Eric Xu, Bradley M. Dickson.

**Project administration:** Parker W. de Waal, Yi Jiang, H. Eric Xu, Bradley M. Dickson.

**Resources:** Parker W. de Waal, Yi Jiang, H. Eric Xu, Bradley M. Dickson.

**Software:** Parker W. de Waal, Bradley M. Dickson.

**Supervision:** Karsten Melcher, Yi Jiang, H. Eric Xu, Bradley M. Dickson.

**Validation:** Parker W. de Waal, Bradley M. Dickson.

**Visualization:** Parker W. de Waal.

**Writing – original draft:** Parker W. de Waal.

**Writing – review & editing:** Parker W. de Waal, Karsten Melcher, Yi Jiang, H. Eric Xu, Bradley M. Dickson.

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
