## [Decision Letter · Decision Letter 0]

2 Oct 2019

Dear Dr de Waal,

Thank you very much for submitting your manuscript 'Molecular mechanisms of fentanyl mediated β-arrestin biased signaling' for review by PLOS Computational Biology. Your manuscript has been fully evaluated by the PLOS Computational Biology editorial team and in this case also by independent peer reviewers. The reviewers appreciated the attention to an important problem, but reviewer 2 raised a number of points about the manuscript as it currently stands. While your manuscript cannot be accepted in its present form, we are willing to consider a revised version in which the issues raised by this reviewer have been adequately addressed. We cannot, of course, promise publication at that time.

Sincerely,

Bert L. de Groot

Associate Editor

PLOS Computational Biology

Nir Ben-Tal

Deputy Editor

PLOS Computational Biology

[LINK]

Reviewer's Responses to Questions

**Comments to the Authors:**

Reviewer #1: This is very important and well written report describing in the first time proper structure-based drug-design of GPCR biased agonists. Such ligands hold high therapeutic potential as being more efficient and clean from side-effects. It is very promising area however very challenging. The key challenge is to isolate the key features in GPCR binding site responsible for G-protein or β-arrestin signalling/activation pathway (via binding of the agonist). The authors used adaptively biased molecular dynamics simulations to predict which chemical features dictate G protein or β-arrestin signalling. The resulting fentanyl-bound pose provides rational insight into a wealth of historical structure-activity-relationship on its chemical scaffold. Authors found that fentanyl and the synthetic opioid peptide DAMGO require M153 to induce βarrestin coupling, while M153 was dispensable for G protein coupling. We propose and validate a mechanism where the n-aniline ring of fentanyl mediates μOR β-arrestin through a novel M153 “microswitch” by synthesizing fentanyl-based derivatives that exhibit complete, clinically desirable, G protein biased coupling.

Such approach is pioneering and it is great to see that it is working. It is very timely report that would be interesting to wide audience of drug-discovery researches and encourage them to apply such approach in their research.

My recommendation will be to publish it as it is.

Reviewer #2: This manuscript reports a sophisticated and massive simulation study coupled with biochemical bench work on the pharmacology of mu-opioid receptor modulators, all the way to proposing new compounds of pharmacological interest. The scientific results are definitely interesting. However, the simulation methodology is oversold (notably by downplaying its cost and requirements), and the writing is of insufficient quality for publication at this stage.

The abstract and author summary give somewhat different accounts of the mABP simulations:

Abstract: "Here we use adaptively biased molecular dynamics simulations to determine how fentanyl (...) activates the μ-opioid receptor"

Author summary: "we use advance [sic] molecular dynamics techniques to obtain the bound geometry of mu-opioid receptor with fentanyl (and derivatives)"

Here the summary is more precise than the abstract, which suggests that the simulations say something about activation, which as far as I know they don't.

About the biasing coordinates: first it is not clear to me what these coordinates are. The text describes reference points that could define center-of-mass distances, but then the coordinates are described as RMSDs, although I cannot see what are the reference coordinates for such an RMSD.

From the Results:

"Importantly, the underlying mABP is blinded to any exogenous information including the crystallographic ligand position"

Judging from Figure S1a and S1b, that statement is implausible, as each group of the ligand is paired with a suitable part of the binding site, which is clear from the PMFs, as the optimal bound pose coincides with the smaller values of both CVs. The conclusion from that is clear: the correct bound pose is, at least in part, encoded in the choice of CVs. It is unclear how simulations with a more naive choice of CVs would fare.

Since the Discussion mention the benefits of this computational approach over virtual screening, it should offer the basis of a fair comparison by stating its total computational cost. The reported mABP simulations cover many tens of microseconds, and the calibration simulations are mentioned but not described in any quantitative detail. What is the total computational cost of the study?

Accurate modeling of the ligand seems essential for this study, and yet, ligand force field parameters from CGenFF were apparently taken at face value without critical evaluation. CGenFF does provide penalties that give feedback on the expected accuracy of the parameters. It would be interesting to read about those. It would seem wasteful to use a sophisticated sampling method to sample a force field of unknown accuracy. A point of detail here: the team beahind CGenFF made the highly unfortunate and confusing choice of calling the parameter assignment program the same name as the force field itself. To decrease the confusion slightly, this manuscript could say the parameters were assigned by the CGenFF *program*.

Fig. 1a's caption is insufficient. One might guess that "30k" etc. indicate the number of atoms in each resepctive benchmark, but that should be spelled out in the figure or caption. The benchmarks should also be documented in the methods section, where there is no mention whatsoever of them.

The Methods state that non-bonded interactions were cut at 8 Angstrom. I assume an approximate long-range LJ term was introduced, this should be stated. Depending on exactly what type of long-range expression is used, it may not be accurate for strongly anisotropic systems like this.

Specific comments

As the manuscript features no page (let alone line) numbers, my references to the text will be approximate.

I started collecting typos and then gave up. Please run this through some form of spell check, and a good proofreading for grammar and syntax.

Author list:

No author is indicated as affiliated to institution #3.

Results:

*"GROAMCS"

*"rotomer", *"rotomeric" (many instances)

Methods:

*"atomospheric" I suppose that means atmospheric, but then it's a strange turn of phrase. Please clarify.

What do the "5000 rounds of refinement" with Modeller refer to specifically?

The text should explain the acronym ICL3 for the benefit of non-GPCR-specialist readers.

The sentence "Substantial pharmacophore overlap..." is missing a verb.

**Have all data underlying the figures and results presented in the manuscript been provided?**

Reviewer #1: Yes

Reviewer #2: Yes

PLOS authors have the option to publish the peer review history of their article (what does this mean?). If published, this will include your full peer review and any attached files.

Reviewer #1: No

Reviewer #2: No

---

## [Decision Letter · Decision Letter 1]

16 Dec 2019

Dear Dr de Waal,

Thank you very much for submitting your manuscript 'Molecular mechanisms of fentanyl mediated β-arrestin biased signaling' for review by PLOS Computational Biology. Your manuscript has been fully evaluated by the PLOS Computational Biology editorial team and the reviewers. Reviewer 2 continues to have reservations about the used methodology and its description, based on which your manuscript cannot be accepted in its present form. We are willing to consider one more revised version in which the issues raised by the reviewers have been adequately addressed. We cannot, of course, promise publication at that time.

Please notice that under these circumstances we might have to consult with a third reviewer.

Sincerely,

Bert L. de Groot

Associate Editor

PLOS Computational Biology

Nir Ben-Tal

Deputy Editor

PLOS Computational Biology

[LINK]

Reviewer's Responses to Questions

**Comments to the Authors:**

Reviewer #2: Attached.

**Have all data underlying the figures and results presented in the manuscript been provided?**

Reviewer #2: Yes

PLOS authors have the option to publish the peer review history of their article (what does this mean?). If published, this will include your full peer review and any attached files.

Reviewer #2: No

---

## [Editor Report · Decision Letter 2]

20 Feb 2020

Dear Mr. de Waal,

We are pleased to inform you that your manuscript 'Molecular mechanisms of fentanyl mediated β-arrestin biased signaling' has been provisionally accepted for publication in PLOS Computational Biology.

Before your manuscript can be formally accepted you will need to complete some formatting changes, which you will receive in a follow up email. A member of our team will be in touch within two working days with a set of requests.

Best regards,

Bert L. de Groot

Associate Editor

PLOS Computational Biology

Nir Ben-Tal

Deputy Editor

PLOS Computational Biology

---

## [Editor Report · Acceptance letter]

2 Apr 2020

PCOMPBIOL-D-19-01436R2 

Molecular mechanisms of fentanyl mediated β-arrestin biased signaling

Dear Dr de Waal,

I am pleased to inform you that your manuscript has been formally accepted for publication in PLOS Computational Biology. Your manuscript is now with our production department and you will be notified of the publication date in due course.

With kind regards,

Sarah Hammond
